🔓 | **Open Peer Review** | Epidemiology | New-Data Letter

# Genomic evidence of Oropouche virus autochthonous circulation in a small district in the state of Rio de Janeiro, Brazil

Filipe Romero Rebello Moreira,[1,2] Fábio Luís Lima Monteiro,[1] Mariane Talon de Menezes,[1] Pedro Junior Pinheiro Mourão,[1] Camila de Almeida Velozo,[1] Matheus Villanueva Andrade,[1] Andréa Cony Cavalcanti,[3] Mário Sérgio Ribeiro,[4] Sílvia Cristina de Carvalho,[4] Claudia Maria Braga de Mello,[4] Rafael Mello Galliez,[2,5] Terezinha Marta Pereira Pinto Castiñeiras,[2,5] Fernanda de Bruycker Nogueira,[6] Carolina Cardoso dos Santos,[6] Ana Maria Bispo de Filippis,[6] Átila Duque Rossi,[1,2] Amilcar Tanuri,[1,2] Carolina Moreira Voloch[1,2]

**KEYWORDS**  genomic epidemiology, phylogeography, emergence, virology

Oropouche virus (OROV; *Peribunyaviridae*), is an emerging arbovirus endemic to the Amazon Basin region of South America, which causes a febrile disease named Oropouche fever (1). This disease is usually mild and presents symptoms similar to other arboviral diseases, but more severe infections have been recently reported, including the first cases of abortion and fetal malformation due to vertical transmission and deaths in 2024 (2–7). This year marked the first OROV outbreak in Brazil outside the Amazon region, following successive resurgences in the endemic area (8). Genomic surveillance indicates that these outbreaks were caused by a reassortant OROV lineage, which presents the S, M, and L segments more closely related to distinct viral strains, including the OROV prototype, Iquitos, and Perdões viruses (9, 10). Preliminary experimental evidence suggests that this lineage replicates faster in mammalian cells and is less effectively neutralized by antibodies from prior infections compared to the prototype OROV strain (11).

Prompted by the identification of several cases of OROV throughout Brazil (12), an effort to monitor the initial OROV cases in the state of Rio de Janeiro (RJ) was employed. The State Health Department of RJ initiated a screening effort employing a standardized molecular diagnostic assay (13), revealing 116 RT-qPCR positive samples scattered across the state, with a considerable cluster of cases ($n = 42$, 36.21%) in the small district of Cacaria (Piraí municipality), which has a population of approximately 800 people (Fig. 1A; Fig. S1). Local surveillance detected the first OROV cases in mid-March 2024, with a subsequent increase in cases over the next five epidemiological weeks, followed by a steady decline through early July (Fig. 1B).

To further investigate this outbreak, we performed virus isolation, whole-genome sequencing, and phylogenetic analysis of four clinical samples (see Text S1). Our analyses confirm that the viruses circulating in RJ belong to the reassortant lineage that spread throughout Brazil in 2024 (Fig. S2). The Bayesian phylogenetic reconstruction from the M data set shows that the sequenced isolates form a monophyletic cluster with strong statistical support (posterior probability = 1), dated to February 2024 (95% highest posterior density: 20 January–2 March 2024), indicating a local transmission chain in RJ (Fig. 1C). A discrete phylogeographic reconstruction suggests that the virus was likely introduced from Minas Gerais in early 2024, though with modest statistical support (posterior probability = 0.64; geographic model posterior probability = 0.43). All four isolates obtained from specimens circulating in Cacaria displayed similar plaque morphology and cytopathic effects (CPE) to the prototypic OROV BeAn strain in culture, even though they belong to the reassortant lineage (Fig. S3). As the prototype OROV

**Peer Reviewer** Emily N. Gallichotte, Colorado State University, Fort Collins, Colorado, USA

Address correspondence to Carolina Moreira Voloch, carolvoloch@gmail.com.

Filipe Romero Rebello Moreira and Fábio Luís Lima Monteiro contributed equally to this article. Author order was determined randomly.

The authors declare no conflict of interest.

See the funding table on p. 4.

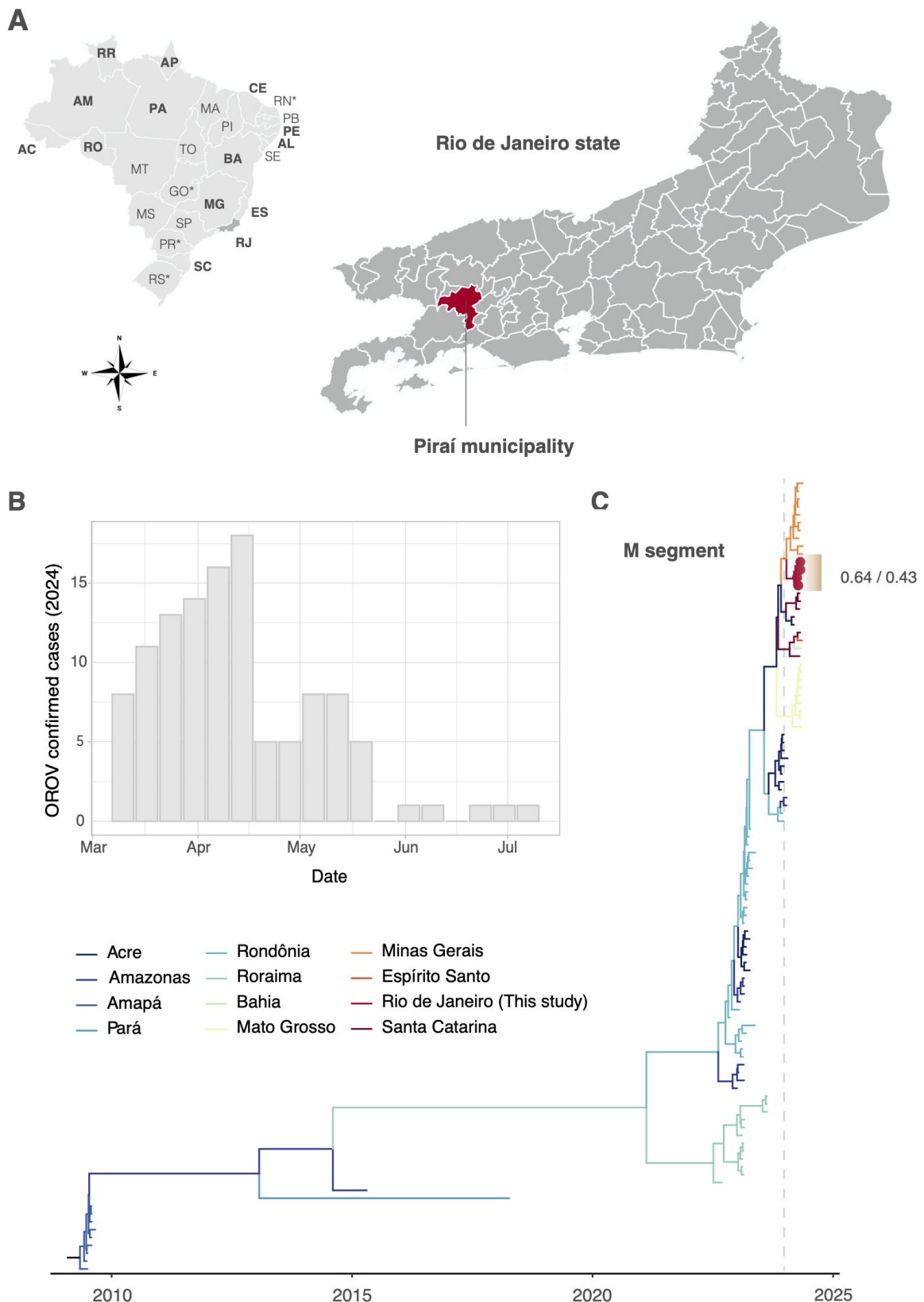

**FIG 1** Epidemiological and phylogenetic contextualization of the Oropouche virus outbreak in Rio de Janeiro state, Brazil. (A) Map of Brazil and Rio de Janeiro state. Two letter codes for Brazilian states were provided, according to Instituto Brasileiro de Geografia e Estatística (https://www.ibge.gov.br/cidades-e-estados, (Continued on next page)

**Fig 1 (Continued)**

last accessed 25 October 2024). States with more than 100 OROV cases in 2024 were highlighted with bold fonts. Asterisks denote states where no case was confirmed. The Piraí municipality, marked in dark red, comprises one-third of OROV laboratory-confirmed cases in RJ. (B) Number of confirmed OROV infections per epidemiological week in RJ. (C) Time-scaled phylogenetic tree of the M segment data set. The model indicates that the sequences cluster in an RJ monophyletic clade, supporting local viral transmission. This clade was dated to the first trimester of 2024 and was most likely introduced from Minas Gerais state. Colors indicate ancestral locations as inferred on a discrete phylogeographic model. Sequences presenting tip shapes (circles) were characterized in this study. Numbers near novel sequences indicate the statistical support for the phylogenetic and trait models, respectively. The dashed gray line marks the beginning of the year 2024.

strain is likely adapted to *in vitro* growth compared to recent isolates, these findings should be interpreted with caution.

This study provides molecular evidence of OROV autochthonous transmission in a non-endemic area. Although the cases were concentrated in Cacaria, this area is linked to larger urban centers, which are well-integrated by road networks to densely populated regions, such as the Rio de Janeiro metropolitan area and the state of São Paulo. Given the local patterns of arbovirus transmission (14), the population's susceptibility to OROV, and climatic conditions favorable to vectors (15, 16), we hypothesize that the virus could spread further across the state, potentially leading to larger epidemics in future transmission seasons. Continuous epidemiological monitoring through molecular, genomic, and serological methods will be critical to assessing the public health impact of OROV circulation in Rio de Janeiro.

## ACKNOWLEDGMENTS

We acknowledge all authors who swiftly submitted Oropouche virus genome sequences to NCBI GenBank during recent outbreaks. We thank public health professionals engaged in generating early data on OROV transmission in Rio de Janeiro state and Piraí municipality.

This study was supported by funding from the Instituto Todos Pela Saúde (ITPS) and Fundação Carlos Chagas Filho de Amparo à Pesquisa do Estado do Rio de Janeiro (FAPERJ, project: E-26/E-26/010/001278/2016). The authors also thank Coordenação de Aperfeiçoamento de Pessoal de Nível Superior (CAPES) and Conselho Nacional de Desenvolvimento Científico e Tecnológico (CNPQ) for the scholarships awarded.

## AUTHOR AFFILIATIONS

[1]Departamento de Genética, Universidade Federal do Rio de Janeiro, Rio de Janeiro, Brazil

[2]Núcleo de Enfrentamento e Estudos de Doenças Infecciosas Emergentes e Reemergentes, Universidade Federal do Rio de Janeiro, Rio de Janeiro, Brazil

[3]Laboratório Central Noel Nutels, LACEN-RJ, Rio de Janeiro, Brazil

[4]Secretaria Estadual de Saúde, SES RJ, Rio de Janeiro, Brazil

[5]Departamento de Doenças Infecciosas e Parasitárias, Faculdade de Medicina, Universidade Federal do Rio de Janeiro, Rio de Janeiro, Brazil

[6]Laboratório de Flavivírus, Fundação Oswaldo Cruz, Rio de Janeiro, Brazil

## AUTHOR ORCIDs

Filipe Romero Rebello Moreira  http://orcid.org/0000-0002-7162-5070
Fábio Luís Lima Monteiro  http://orcid.org/0000-0002-9107-6865
Amilcar Tanuri  http://orcid.org/0000-0003-0570-750X
Carolina Moreira Voloch  http://orcid.org/0000-0001-5182-4366

## FUNDING

| Funder | Grant(s) | Author(s) |
|---|---|---|
| Instituto Todos pela Saúde | | Amilcar Tanuri |
| Fundação Carlos Chagas Filho de Amparo à Pesquisa do Estado do Rio de Janeiro (FAPERJ) | 26/E-26/010/001278/2016 | Amilcar Tanuri |
| Coordenação de Aperfeiçoamento de Pessoal de Nível Superior (CAPES) | | Amilcar Tanuri |
| Conselho Nacional de Desenvolvimento Científico e Tecnológico (CNPq) | | Amilcar Tanuri |

## AUTHOR CONTRIBUTIONS

Filipe Romero Rebello Moreira, Conceptualization, Data curation, Formal analysis, Funding acquisition, Investigation, Methodology, Project administration, Writing – original draft, Writing – review and editing | Fábio Luís Lima Monteiro, Conceptualization, Formal analysis, Investigation, Methodology, Writing – review and editing | Mariane Talon de Menezes, Data curation, Formal analysis, Funding acquisition, Investigation, Methodology | Pedro Junior Pinheiro Mourão, Formal analysis, Investigation, Methodology | Camila de Almeida Velozo, Formal analysis, Funding acquisition, Investigation, Methodology | Matheus Villanueva Andrade, Formal analysis, Investigation | Andréa Cony Cavalcanti, Project administration | Mário Sérgio Ribeiro, Project administration | Sílvia Cristina de Carvalho, Project administration | Claudia Maria Braga de Mello, Project administration | Rafael Mello Galliez, Data curation, Project administration, Resources | Terezinha Marta Pereira Pinto Castiñeiras, Project administration, Supervision | Fernanda de Bruycker Nogueira, Data curation, Project administration | Carolina Cardoso dos Santos, Data curation, Project administration | Ana Maria Bispo de Filippis, Data curation, Project administration | Átila Duque Rossi, Funding acquisition, Project administration, Supervision | Amilcar Tanuri, Conceptualization, Funding acquisition, Methodology, Project administration, Resources, Supervision, Writing – review and editing | Carolina Moreira Voloch, Conceptualization, Data curation, Formal analysis, Funding acquisition, Methodology, Project administration, Resources, Supervision, Writing – review and editing

## DATA AVAILABILITY

All sequences characterized in this study have been deposited in NCBI GenBank under accession numbers PQ537321–PQ537332. These accession numbers are also listed in Table S1.

## ADDITIONAL FILES

The following material is available online.

### Supplemental Material

**Figure S1 (Spectrum02850-24-s0001.tiff).** Map of Cacaria district (State of Rio de Janeiro).
**Figure S2 (Spectrum02850-24-s0002.tiff).** Maximum-likelihood phylogenetic trees inferred from the S, M, and L segment data sets.
**Figure S3 (Spectrum02850-24-s0003.tif).** Virological characteristics of OROV clinical isolates.
**Supplemental Text (Spectrum02850-24-s0004.docx).** Supplemental methods and legends.
**Table S1 (Spectrum02850-24-s0005.xlsx).** Assembly summary statistics and metadata for samples analyzed in this study.

## Open Peer Review

**PEER REVIEW HISTORY (review-history.pdf).** An accounting of the reviewer comments and feedback.

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
