## [Reviewer comments · Microbiology Spectrum]

Microbiology Spectrum

Genomic Evidence of Oropouche Virus Autochthonous Circulation in a Small District in the State of Rio de Janeiro, Brazil

Filipe Moreira, Fábio Monteiro, Mariane de Menezes, Pedro Mourão, Camila Velozo, Matheus Andrade, Andréa Cavalcanti, Mario Ribeiro, Sílvia Cristina de Carvalho, Claudia Maria de Mello, Rafael Galliez, Terezinha Castiñeiras, Fernanda Bruycker Nogueira, Carolina dos Santos, Ana Maria Bispo de Filippis, Átila Rossi, Amílcar Tanuri, and Carolina Voloch

Corresponding Author(s): Carolina Voloch, Universidade Federal do Rio de Janeiro

Review Timeline:

Submission Date:	November 6, 2024
Editorial Decision:	December 9, 2024
Revision Received:	December 18, 2024
Accepted:	January 2, 2025

Editor: Maria Grazia Cusi

Reviewer(s): Disclosure of reviewer identity is with reference to reviewer comments included in decision letter(s). The following individuals involved in review of your submission have agreed to reveal their identity: Emily N Gallichotte (Reviewer #2)

Transaction Report:

DOI: <https://doi.org/10.1128/spectrum.02850-24>

Re: Spectrum02850-24 (Genomic and Virological Evidence of Oropouche Virus Autochthonous Circulation in a Small District in the State of Rio de Janeiro, Brazil)

Dear Dr. Carolina Moreira Voloch:

Thank you for the privilege of reviewing your work. Below you will find my comments, instructions from the Spectrum editorial office, and the reviewer comments.

Revision Guidelines

Sincerely,
Maria Grazia Cusi
Editor
Microbiology Spectrum

Reviewer #1 (Comments for the Author):

This very brief manuscript describes virological and genomic findings from the autochthonous transmission of Oropouche virus infection in a non-endemic area within the state of Rio de Janeiro. The manuscript provides important information about this reemerging arboviral disease.

The authors show results obtained by evaluating data from an outbreak in which within 4 months 36% of the population of a

small community became infected with OROV. The manuscript presented as a New-Data Letter is too short, for better understanding I think it may be useful to transform the manuscript into a structured research article. Please see below my specific comments and suggestions.

Lines 21-23: Please merge and rephrase the first two sentences, Oropouche virus infection causes disease Oropouche fever.

Lines 23-24: The authors should refer to both the documented deaths cases, as well as the abortion and fetal malformations due to vertical transmission. Bibliographic references are not sufficient.

Line 28: What the authors meant by prototypical strains and which of them the authors refer to? Please better specify.

Line 34: State the method used for molecular diagnosis.

Lines 49-51: The results obtained may be affected by the fact that, compared with recent isolates, the prototype strain could be well adapted to in vitro growth; the authors should present these data with caution and should refer to this possibility.

Line 58: Also in this case the authors should pay caution.

Main text references: Please pay attention on the correct indication of bibliographic entries, as required by the journal.

Supplementary text:

The results describing Figure S1 and the reference in the text are missing.

The results for the phylogeny studies shown in Figure S2 are missing. Also missing is the reference to the figure in the text.

Reviewer #2 (Comments for the Author):

see attachment

Minor Comments

1. **Line 35** – 42 of 116 is 36.21% not 36.20%.
2. **Lines 49-51** – States the new isolates “**behave similarly to** the OROV BeAn strain in culture”, however the only comparison between the new isolates and BeAn in figure 3 are CPE and plaque morphology.
 - a. According to the CPE results (SFig 3A), the new isolates look quite different from BeAn CPE, so I disagree they “behave similarly”. The plaque morphology (SFig 3C) does look similar between BeAn and the new isolates though.
 - b. Why is BeAn not included in panel B? The legend states that the plaque images from panel C come from the plaque assay results generated in Figure 3B, so there should be results for BeAn as well. It would therefore be helpful to know how the titers compare between BeAn and the new isolates to better understand if there are in vivo replication differences between the prototypic strain and new isolates.
 - c. Regardless of data provided in B, this section should be updated to more accurately reflect the results “All four isolates.....generate comparable plaque morphology to the prototypic BeAn strain in mammalian cell culture”.
3. **Line 53** – States this study provides “virological and molecular evidence”, however I’m not convinced what the virological evidence is demonstrating autochthonous transmission and should instead just state “molecular evidence”.
4. **Supplementary 1.3** –
 - a. First line of this section states “samples”, but it is unclear if these are the clinical plasma samples, or virus that had been isolated and amplified in the previous section. This should be updated to clearly state which samples are being used. Additionally, it would be helpful to note what method/kit was used to isolate vRNA from samples.
 - b. Then end of the first sentence is missing the word “described”.
5. **Supplemental Figure 3B** – there is a bracket showing “plasma samples”, but it looks as if that is just referring to LVM-3 and LVM-4. It is unclear from the graph and legend, whether these all samples are the actual plasma samples, or virus that has been recovered after amplification on Vero E6 cells. The figure legend and graph legends should be updated to make more clear which samples are being measured, and as noted above, the results from the BeAn strain should be included as well.

Reviewer #1 (Comments for the Author)

Spectrum02850-24 (Genomic and Virological Evidence of Oropouche Virus Autochthonous Circulation in a Small District in the State of Rio de Janeiro, Brazil)

Reviewer #1 (Comments for the Author):

This very brief manuscript describes virological and genomic findings from the autochthonous transmission of Oropouche virus infection in a non-endemic area within the state of Rio de Janeiro. The manuscript provides important information about this reemerging arboviral disease.

The authors show results obtained by evaluating data from an outbreak in which within 4 months 36% of the population of a small community became infected with OROV. The manuscript presented as a New-Data Letter is too short, for better understanding I think it may be useful to transform the manuscript into a structured research article.

R: We appreciate the reviewer's thoughtful suggestion. We chose to present our findings in a new-data letter format due to the nature of the study, which focuses on a concise dataset of novel genetic and experimental findings. To address the reviewer's input, we incorporated all the requested details into the manuscript while retaining the letter format. This approach aligns with our immediate objective to conduct further research and deliver broader genomic and virological analyses that illuminate unexpected aspects of OROV infection and epidemiology.

Please see below my specific comments and suggestions.

Lines 21-23: Please merge and rephrase the first two sentences, Oropouche virus infection causes disease Oropouche fever.

R: We thank the reviewer for the suggestion, the text was adjusted accordingly.

Lines 23-24: The authors should refer to both the documented deaths cases, as well as the abortion and fetal malformations due to vertical transmission. Bibliographic references are not sufficient.

R: We thank the reviewer for helping us to improve our manuscript. We included an array of relevant references in the current version of the manuscript.

Line 28: What the authors meant by prototypical strains and which of them the authors refer to? Please better specify.

R: We were referring to the OROV prototype, and Iquitos and Perdões strains, which are included within the diversity of the species *Orthobunyavirus oropoucheense*. These viruses present genomic segments related to the reassortant lineage circulating in the 2023-2024 epidemics. This information was added to the text.

Line 34: State the method used for molecular diagnosis.

R: We included a reference to the molecular diagnostic assay employed.

Lines 49-51: The results obtained may be affected by the fact that, compared with recent isolates, the prototype strain could be well adapted to in vitro growth; the authors should present these data with caution and should refer to this possibility.

R: We thank the reviewer for the suggestion, we opted for moderating the language and referred specifically to plaque morphology and CPE. We added a comment that these findings should be interpreted with caution, as the prototype strain is likely well adapted to in vitro growth.

Line 58: Also in this case the authors should pay caution.

R: We thank the reviewer for the suggestion. As before, we opted for moderating the language, emphasizing the uncertainty of the occurrence of future local epidemics.

Main text references: Please pay attention on the correct indication of bibliographic entries, as required by the journal.

R: We thank the reviewer for the suggestion. All references were carefully reviewed in the current version of the manuscript.

Supplementary text:

The results describing Figure S1 and the reference in the text are missing.

R: We thank the reviewer for noticing our mistake. We included a brief section describing the map in the supplementary text and also referred to it in the manuscript.

The results for the phylogeny studies shown in Figure S2 are missing. Also missing is the reference to the figure in the text.

R: A section describing the phylogenetic results was added to the supplementary text, as were references to the figure in the main text.

Reviewer #2:

Minor Comments

1. Line 35 – 42 of 116 is 36.21% not 36.20%.

R: The value was corrected in the current version of the manuscript.

2. Lines 49-51 – States the new isolates “behave similarly to the OROV BeAn strain in culture”, however the only comparison between the new isolates and BeAn in figure 3 are CPE and plaque morphology.

R: We acknowledge the reviewer's point that viral “behavior” in cell culture encompasses a complex array of data, including replication kinetics and viral titers, in addition to the CPE morphologic characterization we conducted in our study. Moreover, making such complex comparisons requires the standardization of factors like the multiplicity of infection and other inoculation parameters, which was beyond the scope of our preliminary work. Consequently, we have revised the mentioned sentence to clarify that our study only compared CPE morphology.

a. According to the CPE results (SFig 3A), the new isolates look quite different from BeAn CPE, so I disagree they “behave similarly”. The plaque morphology (SFig 3C) does look similar between BeAn and the new isolates though.

R: When we referred to a similar CPE type observed in Vero E6, our intention was not to compare the intensity of the CPE generated by the OROV isolates and the BeAn 19991 prototypic strain. The intensity of the CPE may be influenced by the amount of viral input and the time post-inoculation, which we could not assess as we had just isolated these viruses from plasma samples. We agree with the reviewer that the chosen micrograph field could lead to misinterpretation of our statement. Therefore, we selected another image from an earlier time point of BeAn 19991 infection in Vero E6 cells and updated Figure S3A and the text was adjusted accordingly.

b. Why is BeAn not included in panel B? The legend states that the plaque images from panel C come from the plaque assay results generated in Figure 3B, so there should be results for BeAn as well. It would therefore be helpful to know how the titers compare between BeAn and the new isolates to better understand if there are in vivo replication differences between the prototypic strain and new isolates.

R: Following the reviewer's comments, comparing two or more viral strains requires detailed analysis and precise experimental settings, such as multiplicity of infection, viral growth in different cell lines, and temporal analysis. In our study, we used the OROV BeAn 19991 strain as a control because of our extensive experience with this virus in culture and the fact that most Oropouche literature relies on this prototypic strain. Given the current Oropouche outbreak context, with the first associated deaths being reported, we must be cautious when comparing OROV strains. The prototypic OROV strain was cultured under very different conditions compared to the novel reassortant OROV isolates. We aim to show that the OROV isolates exhibited high titers shortly after isolation in Vero E6 cells, whereas the prototypic strain was grown under previously established conditions to obtain sufficient titers for in vitro experiments. Therefore, we decided not to include OROV BeAn 19991 titers to avoid any misinterpretation that the emerging OROV reassortant is more virulent or has higher viral fitness than the prototypic strain.

c. Regardless of data provided in B, this section should be updated to more accurately reflect the results "All four isolates.....generate comparable plaque morphology to the prototypic BeAn strain in mammalian cell culture".

R: We appreciate the Reviewer's suggestion and updated the text.

3. Line 53 – States this study provides "virological and molecular evidence", however I'm not convinced what the virological evidence is demonstrating autochthonous transmission and should instead just state "molecular evidence".

R: We agree with the reviewer's assessment, the title was adjusted accordingly.

4. Supplementary 1.3 –

a. First line of this section states "samples", but it is unclear if these are the clinical plasma samples, or virus that had been isolated and amplified in the previous section. This should be updated to clearly state which samples are being used. Additionally, it would be helpful to note what method/kit was used to isolate vRNA from samples.

R: We thank the reviewer for their suggestion. For clarity, we changed the word "samples" for "Isolated viruses", more precisely reflecting our workflow. We also included information on the RNA extraction method used.

b. Then end of the first sentence is missing the word "described".

R: We thank the reviewer for noticing our mistake, the word “described” was included.

5. Supplemental Figure 3B – there is a bracket showing “plasma samples”, but it looks as if that is just referring to LVM-3 and LVM-4. It is unclear from the graph and legend, whether these all samples are the actual plasma samples, or virus that has been recovered after amplification on Vero E6 cells. The figure legend and graph legends should be updated to make more clear which samples are being measured, and as noted above, the results from the BeAn strain should be included as well.

We agree with the reviewer's observation that the label position could lead to misinterpretation. We have also updated the legend to emphasize that the viral titer was obtained after viral isolation from the supernatant of Vero E6 cells following CPE detection.

Re: Spectrum02850-24R1 (Genomic Evidence of Oropouche Virus Autochthonous Circulation in a Small District in the State of Rio de Janeiro, Brazil)

Dear Dr. Carolina Moreira Voloch:

I suggest to modify the sentence: 'As the prototype OROV strain is likely adapted to in vitro growth compared to recent isolates.....with caution', because your isolates grew at high titer in cells. This indicates that these recent strains do not need to be adapted to in vitro growth.

Your manuscript has been accepted, and I am forwarding it to the ASM production staff for publication. Your paper will first be checked to make sure all elements meet the technical requirements. ASM staff will contact you if anything needs to be revised before copyediting and production can begin. Otherwise, you will be notified when your proofs are ready to be viewed.

Sincerely,
Maria Grazia Cusi
Editor
Microbiology Spectrum